# Protein Phase Separation: New Insights into Carcinogenesis

**DOI:** 10.3390/cancers14235971

**Published:** 2022-12-02

**Authors:** Yan Luo, Shasha Xiang, Jianbo Feng

**Affiliations:** Cancer Research Institute, The First Affiliated Hospital, Hengyang Medical School, University of South China, Hengyang 421001, China

**Keywords:** phase separation, membraneless organelle, biomolecular condensates, oncogenic mutation, oncology therapy

## Abstract

**Simple Summary:**

The global burden of cancer is substantial and growing. Among 22 groups of diseases and injuries in the GBD 2019 study, cancer was second only to cardiovascular diseases for the number of deaths and years of life lost. There are no good therapeutic drugs to deal with tumors in clinical practice. This paper reviews the potential of phase separation as a new field of cancer research in addressing the unmet clinical needs of patients with cancer.

**Abstract:**

Phase separation is now acknowledged as an essential biologic mechanism wherein distinct activated molecules assemble into a different phase from the surrounding constituents of a cell. Condensates formed by phase separation play an essential role in the life activities of various organisms under normal physiological conditions, including the advanced structure and regulation of chromatin, autophagic degradation of incorrectly folded or unneeded proteins, and regulation of the actin cytoskeleton. During malignant transformation, abnormally altered condensate assemblies are often associated with the abnormal activation of oncogenes or inactivation of tumor suppressors, resulting in the promotion of the carcinogenic process. Thus, understanding the role of phase separation in various biological evolutionary processes will provide new ideas for the development of drugs targeting specific condensates, which is expected to be an effective cancer therapy strategy. However, the relationship between phase separation and cancer has not been fully elucidated. In this review, we mainly summarize the main processes and characteristics of phase separation and the main methods for detecting phase separation. In addition, we summarize the cancer proteins and signaling pathways involved in phase separation and discuss their promising future applications in addressing the unmet clinical therapeutic needs of people with cancer. Finally, we explain the means of targeted phase separation and cancer treatment.

## 1. Introduction

Biological evolution is the evolutionary process of the development of all life forms, and an important characteristic is the progressive complexity and refinement of the different levels of morphological structures. Typically, different biological reactions take place in different organelles in an orderly manner; for example, transcription occurs primarily in the nucleus, protein modification occurs in the endoplasmic reticulum and Golgi, and molecular degradation occurs in lysosomes. These organelles are surrounded by a single or double layer of molecular membranes and are isolated from the surrounding environment. To ensure that various cellular components can aggregate at the correct time and in the proper space to perform their corresponding functions, cellular molecules are isolated in different cellular compartments as needed. In addition to classic membrane-bound organelles, much evidence further suggests that cells possess various membraneless compartments, including the nucleolus, Cajal body, and PcG body in the cellular nucleus [1], along with stress granules (SGs) and P-bodies in the cytoplasm [2,3,4,5]. Proteins, RNA, and other molecular components constitute these membraneless organelles, and phase separation drives the processes involved [2,6]. Although not covered by cell membranes, membraneless organelles are still capable of frequent molecular exchange with the environment. With ongoing scientific inquiry, many insights have been gained into the structure of these organelles. However, questions about why these organelles form, the mechanisms involved, and how their biological properties affect their function have not been answered. As science has advanced, these questions have begun to be resolved, and a deeper understanding of the organization, molecular properties, and regulation of membraneless organelles has emerged [7,8]. In recent years, there has been growing evidence that membraneless organelles are involved in the development of multiple cancers [9,10]. These findings have given rise to a new domain of cellular biology. The emphasis is on learning how cell substances are organized into membraneless organelles, how to promote their activity, and how disorders in these organelles frequently lead to diseases, encouraging researchers to consider the biological processes involved from the viewpoint of phase separation.

Cancer is a disease that seriously threatens human health. Both in terms of morbidity and mortality, cancer is a major global public health problem and second only to cardiovascular disease in terms of mortality [11]. Although research on the occurrence and development of cancer has a history spanning many years, its treatment is still a great problem facing the world. To overcome this challenge, new concepts are urgently needed to characterize and explain the complicated mechanisms of human cancer. An increasing body of evidence suggests that there is a close link between abnormal phase separation condensation assembly and aberrant oncogenic procedures. The advent of protein phase separation offers a novel possibility for targeting refractory cancer.

In this review, we summarize the formation, adjustment, and function of phase separation, focusing on the mechanism of phase separation and various phase separation-related proteins in the course of tumor progression. We also emphasize their emerging role in cancer pathogenesis, as well as new approaches to cancer treatment, which offer a useful basis for the eventual treatment of complicated human cancers. In summary, this review will help researchers understand phase separation during cancer development and fully realize the potential of targeting phase separation in oncotherapy.

## 2. Overview of Phase Separation

Phase separation is a process by which a well-mixed solution of macromolecules such as proteins or nucleic acids spontaneously separates into two phases: a dense phase and a dilute phase [12]. Whether phase separation occurs in a solution depends greatly on the concentration and properties of the macromolecules and the solution, as well as on the environmental conditions [13]. With a growing understanding of the basic molecular principles of phase separation, there is an awareness of the different functions of phase separation in various cellular processes. In general, the main ones include stress response, regulation of gene expression and control of signal transduction [6,14], protein degradation [15], cytoskeleton assembly [16], and gene activation [17] or repression [18], including epigenetics, transcription, and translation. Furthermore, it has been demonstrated that many fundamental biological processes are inseparable from phase separation, including heterochromatin formation [18,19], nucleocytoplasmic transport [20], supramolecular assemblies [21], and assembly of membraneless compartments, such as SGs [22]. Cell architectures formed by phase separation are named biomolecular condensates to mirror their provenance through condensation reactions [2,23]. Contrary to other types of components, they can enrich molecules, and rapid exchange of components and agglomeration of droplets can form specific cellular structures called membraneless organelles by phase separation. In different physiopathological situations, biomolecular condensates can be converted into different states of matter. Similarly, condensates play an essential role in the life activities of various organisms, including advanced structures, gene expression regulation [24], autophagic degradation of incorrectly folded or unneeded proteins [25], signal cluster assembly, and synaptic plasticity regulation of the formation of signaling molecules [10].

### 2.1. Phase Separation to Form Membraneless Organelles

In 2009, Hyman and Brangwynne found that some properties of P granules resemble liquids and that regulated dissolution/condensation drives their localization, and the researchers first realized that membraneless organelles may be driven by phase separation [26]. Membraneless organelles are dynamic structures with liquid-like physical properties [26]. Due to a lack of lipid-rich membranes, changes in the surrounding environment can easily affect their internal homeostasis, so the protein composition and morphology of membraneless organelles respond accordingly to changes in the cellular environment, and this ability may represent the mechanism underlying the stress response sensed by membraneless organelles [27]. For example, oncogenic ARF protein is localized in the nucleolus and released into the nucleoplasm through changes in phase separation in response to environmental stresses of DNA damage and oncogene activation, hence activating the p53 oncogenic pathway [28].

What is the unique role of membraneless organelles driven by phase separation? The nucleolus is the largest and most intensively studied membraneless organelle, serving as the center of ribosomal RNA synthesis and nascent ribosome assembly in eukaryotic cells. Ribosome biogenesis is vectorial, starting from fibrillar centers (FCs), where rDNA is transcribed into rRNA [29]. Paraspeckles are located in chromatin gaps and are subnuclear bodies built on NEAT1 (a long noncoding RNA). Paraspeckles are involved in many physiological processes, including the cellular stress response, cell differentiation, corpus luteum formation, and cancer development [30]. The proteins that comprise paraspeckles are related to RNA polymerase II transcription and RNA processing. Cajal bodies (CBs) and histone locus bodies (HLBs) could be responsive to stress conditions, and they are nuclear bodies (NBs) involved in the transcriptional and posttranscriptional regulation of small nuclear RNAs and histone genes [31]. With the deepening of research, an increasing number of membraneless organelles and their functions are being discovered (Figure 1).

### 2.2. Multivalent Interactions Promote the Formation of a Phase Separation Network

Recent studies have shown that biomolecular phase separation occurs through multivalency or the capacity to participate in weak multivalent effects that rapidly assemble, disconnect, and recombine. These multiple interactions are facilitated by proteins that embrace multiple folding module domains or intrinsically disordered regions (IDRs) [32] or oligomerization domains [33]. Another type of phase separation protein contains polymeric structural fields, such as DIX domains, which can cross-link to form three-dimensional condensates [34]. These multivalent interactions mainly include two types: one class of intracellular interactions, such as protein–protein, protein–RNA, and RNA–RNA interactions; and another class of weak, instant, multivalent interplay between IDRs, consisting of π–π interactions, cation–anion interactions, dipole–dipole interactions, and π–cation interactions [10,35,36].

In addition, another mechanism, called bridging-induced phase separation (BIPS), has been demonstrated in a number of chromatin-related phase separation phenomena. It has been confirmed that BIPS is the basis for DNA-mediated clustering cohesion [37]. Local bridging along distal segments of DNA molecules is an essential element of BIPS and a characteristic of BIPS that does not exist in other forms of phase separation [38]. In this way, phase separation is induced between chromatin regions (stable or transient) that interact with different types of bridging factors [38].

How do cells use phase separation to respond to changes in cell surroundings? Cells construct and regulate dynamic membraneless organelles through characteristics encoded in intrinsically disordered proteins (IDPs) of related proteins, many of which play central functions in a variety of cell features. IDPs are conformationally flexible, often interacting with their bonding companions via short sequence motifs that reappear within disordered areas, and this multivalent interplay is common in macromolecular complexes [39]. As T-cell receptor components form a cluster of membrane-associated phase separation signals, during T-cell activation, phase separation is driven by the multivalence of LAT, GRB2, SLP-76, Nck, and WASP [40].

## 3. Mechanisms of Phase Separation in Tumorigenesis

Incorrect or abnormal phase separation of biological macromolecules is closely related to the occurrence of many types of diseases, such as cancer, neurodegenerative diseases, and infectious diseases [41,42,43]. The formation and regulation of aberrant biomolecular condensates is changing the way people think about coping with many diseases, including tumor genesis, diagnosis, and treatment. Researchers are no longer considering highly recurrent point mutations in tumors merely based on the structural visual field but are also considering the condensates involved in these mutations. Generally, combined with current existing research, there are three mechanisms of abnormal biological phase separation leading to disease: (a) The first mechanism is biomacromolecule condensate dysregulation. In cancer, IDR-related signal receptor mutations or chromosome translocations can promote the shape of signal clusters or condensates at transcription or DNA damage repair sites and then change the cell signal cascade, drive abnormal transcription programs or DNA damage repair, and promote cancer cell proliferation [44]. (b) The second mechanism is the changing of the critical catalysts required for phase separation. There is evidence that enzymes can regulate the assembly of biomolecular condensates through posttranslational modification (PTM). For example, DYRK3 is located in condensate and phosphorylates several serine and threonine residue groups in IDRs [45]. During stress recovery, inactivation and activation of DYRK3 are crucial to the formation and dissolution of SG [46]. (c) The third mechanism is the altering of general physicochemical conditions in cells. Cells exposed to stress undergo extreme fluctuations in the levels of ion concentration, osmotic conditions, and pH values, which can change the solubility and interactions of biological macromolecules, resulting in abnormal phase separation [47] (Figure 2).

Cancer is the abnormal proliferation of cells in local tissues under the action of various tumorigenic factors in the body. In addition to having unlimited proliferation and multidirectional differentiation potential, many cancer cells have the ability to evade growth repression, engage in replicative immortality, avoid immune destruction, and cause instability of the genome [48], as well as newly discovered tumor features, such as unlocking phenotypic plasticity and reprogramming nonmutational epigenetics [49]. It is worth noting that gene mutations, tumor-promoting inflammation, unlocking phenotypic plasticity, and polymorphic microbiomes increase the possibility of tumors. Gene mutations in cancer often lead to oncogene activity imbalance or inactivation of tumor suppressor genes, thus promoting the carcinogenic process. Despite considerable advances in our understanding of how mutations promote the carcinogenic process, the exact pathogenesis of tumorigenesis remains unclear, as does the mechanism by which tumor cells acquire these features. Phase separation offers a new direction for understanding cancer phenotypes (Figure 3). Generally speaking, it can be divided into two aspects. Dysregulation is driven by phase separation itself. For example, interferon-γ improves tumor sensitivity to immunotherapy by inhibiting YAP phase separation [50]; phase separation of YAP and TAZ participates in activating EMT [51]; and increasing the formation of SGs overcomes stress-induced cancer cell death [52]. In addition to this, carcinogenicity can also be affected by the dysregulation of signaling proteins involved in phase separation. For instance, MYC forms transcription condensates by binding to superenhancers, which lead to VEGF expression and the promotion of angiogenesis [17]. In addition, aberrant phase segregation of the ENL protein can recruit a large number of associated transcription complexes, which lead to genomic rearrangements in cancer [53]. Fusion between promyelocytic leukemia protein (PML) bodies permit telomere lengthening and enable replicative immortality [54], and SPOP mutation inhibits the catabolism of prooncogenic substrates, thus escaping growth inhibition [55]. An increasing number of studies has shown that the process of phase separation cannot be ignored for the progression and treatment of human diseases.

Abnormality of phase separation may promote the occurrence of some cancers. For example, one study has directly linked protein phase separation to cancer. In vitro, substrates can trigger phase separation of speckle-type BTB/POZ protein (SPOP) and colocalization in membraneless organelles in cells, and carcinogenic mutations in cancer suppressor SPOP result from interference with phase separation and colocalization in membraneless organelles linked to specific phase separation defects [55]. Cullin3-ring ubiquitin ligase is associated with a variety of solid tumors, and SPOP as its substrate adapter is one of the first cancer-specific related proteins to undergo phase isolation [55,56]. Molecular pathologist Miguel Rivera found a protein associated with Ewing sarcoma. This protein can activate oncogene expression when it accumulates near the genome related to tumorigenesis, and abnormal “phase separation” may promote the aggregation of this protein near these regions, leading to the occurrence of Ewing sarcoma [57]. Moreover, the FUS/EWS/TAF15 (FET) fusion oncoprotein enhances abnormal gene transcription by site-specific phase separation and is an indispensable carcinogenic driver in various human cancers [58,59]. This study reveals that phosphatase protein can undergo phase separation, suggesting that phase separation is a notable means for cells to regulate phosphatase activity. Gene mutation can change the phase separation ability of protein and then change the protein function, leading to the occurrence of human diseases, highlighting the importance of phase separation in human disease occurrence and development [59]. The following is a summary of the various types of cancer condensate formation and the regulation of cancer-associated proteins (Table 1 and Figure 4).

## 4. Abnormal Biomolecular Condensates in Cancer-Related Signaling Pathways

Most cancer signature processes involve biomolecular condensates [48]. Multiple signaling pathways that control cell growth, proliferation, migration, and invasion are modified in cancer, and the overexpression, mutation, or fusion of signaling proteins can lead to insufficient activation or overactivation of pathways [120,121,122]. Proteins involved in related pathways have been shown to form biomolecular condensates by phase separation to facilitate signal transduction and ensure fine regulation of downstream signaling pathways (Table 2) [14,123].

### 4.1. cAMP/PKA

In the PKA type I regulatory subunit, RIα vesicles, as key components of the cAMP signaling pathway, experience phase separation to form biomolecular condensates rich in cAMP and PKA activity that control local signal transduction. Abnormal phase separation increases oncogenesis; for example, the PKA fusion oncoprotein associated with atypical hepatocellular carcinoma effectively blocks the phase separation of RIα and induces aberrant cAMP signaling, thereby promoting cell proliferation and inducing cell transformation [124].

### 4.2. cGAS/STING

The tumor suppressor NF2/Merlin, localized in the cell membrane, tight junctions, and cytoskeleton, also mediates intercellular signaling. Mutations and deletions of the NF2 gene are a main cause of the development of neurofibromatosis type 2, while missense mutations can lead to malignant tumors in multiple organs, such as meningiomas [131]. Mechanistically, NF2 mutations or deletions lead to tumorigenesis, mainly due to the aberrant regulation of cell proliferation signaling pathways. Surprisingly, recent studies have revealed that NF2 mutations are also capable of regulating tumor immunity. A point mutant of TEAD converts NF2 into a potent suppressor of cGAS/STING signaling, strongly inhibiting cGAS/STING-mediated cell-autonomous and nonautonomous tumor immunity and suppressing nucleic acid recognition. Nucleic acid recognition, through activation of IRF3, is able to induce protein phase separation and the formation of membraneless organelles and can detect NF2m-IRF3 condensates in patients with neurofibromatosis type 2 [125]. Overall, induction of mutant NF2 strongly suppresses STING-initiated antitumor immunity through phase separation, thereby promoting tumorigenesis.

### 4.3. cEGFR/RAS

Erythroblastic oncogene B (ErbB) belongs to the EGFR family, a subfamily I of the RTK superfamily. Activated ErbB receptors bind to a variety of signal transduction proteins and excite the activation of multiple signal transduction pathways [132]. Epidemiological data show that the overexpression of ErbB receptors in many human tumor tissues is closely related to tumor growth, invasion, and metastasis. ErbB receptors, particularly EGFR and ErbB2, have become preferred targets for cancer therapy. Among all RTKs, EGFR was the first to be discovered and the first to be associated with tumors. EGFR is expressed on the surface of normal epithelial cells, while it is often overexpressed in some tumor cells and is associated with metastasis, invasion, and poor prognosis of tumor cells [132]. There are two major signaling pathways downstream of EGFR: the Ras/Raf/MEK/ERK-MAPK pathway and the PI3K/Akt/mTOR pathway [133]. In addition, the aberrant activation of downstream RAS often leads to abnormal proliferation, survival, and migration of tumor cells [126]. RAS activation occurs when ligand-bound membrane receptors recruit and modify adaptor proteins, and these adaptor proteins form condensates at the cell membrane that compartmentalize proteins, increasing their “dwell time” and RAS activity [59,134]. Furthermore, EGFR activation was found to be related to membrane-bound clusters. Such EGF-dependent formation of EGFR nanoclusters may add an additional spatial regulatory layer to growth factor signaling, which is consistent with the emerging view of how Ras regulates downstream pathways through the formation of similar higher-order species [127]. Therefore, EGFR condensate formation is likely to regulate the protumor genic activation of RAS.

### 4.4. Wnt/β-Catenin

Altered signaling pathways and associated genes that regulate cell growth, proliferation, apoptosis, and invasion are general features of neoplasm, but the degree and mechanisms by which these pathways are altered occur differently among tumors and tumor types [121]. Thus, pathway mutations are associated with many human cancers. During animal development, Wnt signaling regulates the fundamental fate of many cells and plays a substantial role in tissue homeostasis. Aberrant Wnt signaling is observed in many cancer entities, such as colorectal and breast cancers, where it promotes tumor progression by influencing and maintaining cancer stemness, proliferation, and metastasis [122]. It has been shown that the destruction complex plays a role as a biomolecular condensate in the adjustment of the Wnt pathway [128]. In the absence of Wnt stimulation, the destruction complex establishes a scaffold for kinase and β-catenin through the axial protein, phosphorylating β-catenin and promoting its degradation [135]. In the presence of Wnt, the downstream Disheveled (Dvl) protein is activated, and its PDZ structural domain binds to the intracellular carboxyl terminus of the FZD protein [34,136], which triggers downstream signal transduction and enhances the phosphorylation and inhibition of GSK3β, thereby destabilizing the damage complex and negatively regulating the degradation of β-catenin [137].

### 4.5. RAS/MAPK

SG is a remarkable type of RNP granule, a “membraneless organelle” formed in the cytoplasm of eukaryotic cells in response to multiple stressors. It has been demonstrated that SGs consist of unevenly distributed interactions in a network of core protein–RNA interactions, which are then assembled by phase separation to promote cell survival under stress conditions. The assembly of SGs is further regulated by positive or negative synergistic effects of the central node of the network, G3BP1 binding factor [22]. Because tumor cells require a large energy and material base for proliferation and are dependent on various metabolic pathways, cancer cells commonly reside in a distinctive microenvironment featuring irregular vascularization and an inadequate supply of oxygen and nutrients [138]. These conditions stimulate the cellular stress response and induce the aggregation of cytoplasmic SGs, the assembly of which is a major adaptive defense mechanism that promotes the adaptation of cancer cells to the unfavorable microenvironment and enhances the resistance of cancer cells to apoptosis by inhibiting the MAPK pathway of the stress response [129,139].

The nonreceptor protein tyrosine phosphatase SHP2 plays a key role in the RAS/MAPK signaling pathway, and SHP2 phosphatase activity is controlled by its own conformational changes. Mutations in SHP2 protein are closely connected to multiple human diseases, and mutations at different sites can advance abnormal phase separation of SHP2 in vitro and in cells. SHP2 mutant phase segregation forms a protein condensate that recruits wild-type SHP2 and greatly promotes SHP2 phosphatase activity, thereby activating the MAPK signaling pathway [59].

### 4.6. Hippo/YAP

The center of the Hippo pathway includes the kinase cascade, DNA binding partners, downstream effectors, and transcriptional coactivators YAP and TAZ [140,141]. Aberrant activation of YAP is closely associated with cancer development and metastasis [142]. In cells, YAP aggregates into condensates in the superenhancer region, and these nuclear condensates comprise TAZ as well as the transcription factor TEAD1 [61]. Further studies found that YAP, as a major effector downstream of the Hippo signaling pathway, plays a critical role in adjusting organ size, promoting tissue regeneration and cell fate determination [140]. Recently, it was found that SRC-1 is an important coactivator in the YAP/TEAD transcriptional complex, which can enhance the transcriptional activity of YAP by upregulating the downstream genes of YAP. The steroid receptor coactivator 1 (SRC-1) protein has two prion-like domain (PrD) segments that can phase separate and distribute into the YAP/TEAD transcriptional condensate, and, moreover, the SRC-1/YAP/TEAD colocalized puncta are enriched with active transcriptional markers [130].

Furthermore, the strength of condensate assembly is closely connected to the transcriptional activity of YAP and TAZ, which is reduced when YAP and TAZ mutants are formed [51,61]. Upon activation, YAP/TAZ translocate to the nucleus and assemble nuclear condensates. Hippo signaling negatively regulates this process and can respond to multiple mechanical changes in vivo. Consistent with this association, there is growing evidence that YAP and TAZ can act as sensors or checkpoints for cellular mechanical stress, regulating a variety of genes by binding and activating TEAD transcription factors [143]. In this way, YAP and TAZ can act as signaling centers in the tumor microenvironment. For example, YAP and TAZ are key to triggering multiple cell-autonomous responses such as permanent proliferation, cell plasticity, drug resistance, and metastasis. In cancer cells, YAP and TAZ can synergize with stromal cells to sense changes in the surrounding microenvironment, and this mechanical signaling from outside to inside usually activates several oncogenes and transcriptional programs to initiate and maintain oncogenic transformation [96].

In a separate study, excessive glycogen accumulation was found to be prevalent in early liver tumor lesions and small tumors, suggesting that cancer cells in early cancer initiation may draw glucose and store it intracellularly more as glycogen storage rather than metabolize and break down glucose in the form of anaerobic glycolysis. Further experiments revealed that excessive glycogen accumulation leads to liquid–liquid phase separation, driving assembly of the Laforin-MST1/2 complex, causing inactivation of the Hippo pathway that inhibits cell carcinogenesis, and increasing the activity of downstream proto-oncoprotein YAP, which increases hepatocarcinogenesis [60].

### 4.7. NRF2/NF-κB

In addition to the above pathways, NF-κB and NRF2 signaling pathways have been shown to be involved in the regulation of tumor progression by phase separation. The accumulation of stent protein p62 has been shown to activate NRF2, which accelerates hepatocellular carcinoma progression [144] and excites NF-κB and NRF2 signaling to stimulate pancreatic tumor advance [70].

p62 is a ubiquitin-binding autophagy receptor and signaling protein [144]. Recent research found that p62 phase separation is dependent upon p62 polymerization, p62 ubiquitin interaction, and the valence state of the ubiquitin chain. Ubiquitin chains and cytoplasmic mixing of recombinant p62 with p62–/–cells induce p62 phase separation and assembly into the condensate p62 bodies, and the formed condensate is subsequently phagocytosed and degraded by autophagosomes [64,65,145]. Another study found that p62, along with mutant KEAP1 protein and transcription factor NRF2, assembled to form cyclic, insoluble clusters that are phase separation biomolecular condensates, thereby affecting NRF2-driven transcription [65]. Overall, the above findings suggest that p62 condensates are involved in the formation of various cellular condensates and autophagy-mediated disposition that promote or inhibit tumorigenesis.

## 5. Targeting Phase Separation: A New Treatment Option for Tumors

Phase separation is one of the processes of life. Phase separation occurs continuously in life, and it is one of the most important processes in proteins. Its discovery provides a new perspective and concept for the process of cell fate determination and regulation of the expression of key genes during disease development, and the condensates formed by abnormal phase separation can cause cancer. Considering that condensate formation is a phenotype that is relatively easy to visualize by imaging techniques, the rapid identification of drugs that alter condensates appears to be achievable. We can be optimistic that this development will help identify new therapeutic approaches. Target molecules common to tumor research can be identified, providing new ideas for tumor treatment from the perspective of protein phase separation.

### 5.1. Targeted Therapy for Undruggable Proteins

Despite tremendous advances in modern molecular biology in medicine, a large number of disease-associated proteins remain “undruggable”. The emergence of phase separation provides a new way to target refractory and undruggable proteins. For example, SRC-1 is an important coactivator in YAP/TEAD transcriptional complex, and the expression of SRC-1 is positively correlated with the transcriptional activity of YAP/TEAD. Therefore, development of small-molecule compounds that regulate the phase separation of SRC-1 could provide new ideas for the inhibition of YAP signaling in cancer. It has been recently reported that the anti-HIV drug etigevir (EVG/elvitagravir) effectively inhibited cancer-associated phase separation of SRC-1, a nuclear hormone receptor transcriptional coactivator, without affecting YAP/TEAD transcriptional condensates; EVG directly binds to SRC-1 protein and inhibits YAP downstream gene expression and SRC-1 and the YAP-mediated growth of tumor cells [130]. This work presents a phase-separation-based pharmacological strategy to suppress YAP-mediated tumorigenesis by targeting the nondrug SRC-1/YAP/TEAD transcriptional complex, demonstrating the advantages and potential of phase separation as a novel approach to treating nondrug targets and previously refractory diseases.

Likewise, disease-associated SHP2 mutant phase separation was found to be characteristically reduced by SHP2 heterodimeric (allosteric) inhibitors. The nonreceptor tyrosine phosphatase SHP2 plays a significant role in RAS/MAPK signaling, and SHP2 mutants promote excessive activation of the MAPK signaling pathway through aberrant phase segregation. SHP2 phosphatase activity is regulated by its own conformational changes, and conformational changes in SHP2 mutants directly affect their phase separation ability. The SHP2 conformational change inhibitor ET070 was found to specifically inhibit the phase separation ability of SHP2 mutants by locking SHP2 in the “off” conformation [59]. This is the first discovery that phosphatases have the ability to phase separate, implying that protein phase separation may be an important way for cells to regulate phosphatase activity, revealing that gene mutations can cause different human diseases by directly altering protein phase separation ability, further solidifying the key role of protein phase separation disorders in major human diseases and providing a novel strategy for treating relevant major human diseases through small-molecule inhibition of protein phase separation disorders.

### 5.2. A New Pathway for Drug Transport and Delivery

Biomolecules, with critical advantages such as high potency, specificity, and safety, are expected to play a valuable role in the treatment of different diseases. However, their low cell membrane permeability limits their access to intracellular targets. Traditional strategies usually employ nanoscale carriers, which can be isolated in compartments of the intranuclear body. Recently, it has been shown that micron-scale polypeptide clusters prepared by phase separation can cross the cell membrane via a pathway that does not rely on classical endocytosis, where they undergo glutathione-induced payload release, and many macromolecules can be rapidly recruited into microdroplets, including small peptides, enzymes, and mRNAs [146]. Among them are proteins and low-molecular-weight compounds that can be loaded within polysomes, which are expected to provide new pathways for intracellular transport of therapeutics, including cancer and metabolic diseases, or as vaccine carriers on the basis of mRNAs.

Due to the phenomenon of phase separation, anticancer drugs form droplets that accumulate at specific locations in the cell [112]. Exploiting this phenomenon allows certain drugs to be targeted more effectually while limiting nonanticipatory toxicity, which causes deleterious side reactions. This effect generally arises in native molecules, but in a recent study, it was found that synthetic compounds can also be optionally isolated in a similar way in condensates. When cisplatin is mixed with a known cohesion in the nucleus, for example, cisplatin selectively accumulates in a condensate formed by a gene activating a protein called Mediator Complex Subunit 1 (MED1), and the concentration of the drug inside the condensate is 600 times higher than that outside. In addition, since MED1 acts primarily on oncogene promoters, cisplatin finally targets the corresponding DNA with its toxic platinum atoms, in essence attacking the most critical parts of the cancer cells [112].

Founded in 2019, Dewpoint Therapeutics is the first company to publicly stake a claim in drug discovery based on phase separation. In a 2020 study, they found that the phenomenon of phase separation of anticancer drugs also appeared to affect drug resistance. Tamoxifen, a drug used to treat breast cancer, also accumulates in MED1 condensates; however, cancer cells resistant to tamoxifen develop much higher levels of MED1, which may cause condensates to swell and dilute the drug, thus reducing its efficacy [112]. Thus, the condensates formed by phase separation help to optimize the drug distribution, target contact, and therapeutic index of small-molecule drugs.

### 5.3. State Translation of Protein Condensates

In addition to drug aggregation, many proteins also aggregate and cause serious human disease. Of particular note is the aggregation of mutants of the oncogene p53, which is present in more than 50% of malignancies. Under normal conditions, wild-type p53 acts as a tumor suppressor, regulating the expression of its downstream genes to inhibit tumor development. However, p53 exists as a mutant in many tumor cells, forming amyloid aggregates in the nucleus and inducing and promoting tumorigenesis and progression. P53 phase separation and aggregation processes and the regulation of phase separation have been investigated by the team of Jerson L. Silva.

It is believed that the phase separation process of p53 precedes the formation of its aggregates, and that the phase separation process of p53 and its transformation to condensates may be correlated with the oncogenic activity of p53 and may be an essential treatment target. The capability to recognize and block p53 aggregate precursor states affords a novel technique for tracking p53 aggregation and testing the effects of therapies aimed at stabilizing the native form of p53 [147]. However, the phase separation of p53 is closely linked to normal physiological function. Its normal physiological function reversible phase segregation behavior and pathogenic changes in the regulation between irreversible changes in the solid state will be an important issue for relevant drug research, and precise regulation of the phase separation behavior of p53 may become a promising cancer treatment [147].

In addition, the nuclear protein A-kinase anchoring protein 95 (AKAP95), which plays an important role in supporting oncogenesis by splicing control, also forms phase separation and similar liquid condensates in the nucleus [148]. It was shown that mutations in different amino acid key residues interfere with AKAP95 condensation in reverse orientation. Notably, the splicing regulatory activity of AKAP95 is dramatically diminished or even lost with condensate hardening or the disruption of condensation and is restored by replacing its condensation regulatory domain with that of other unrelated proteins. Furthermore, the ability of AKAP95 to regulate gene expression and support tumorigenesis requires AKAP95 to form condensates with appropriate mobility and dynamics [148]. The connection between the phase separation of AKAP95 and tumorigenesis may provide chances for cancer therapeutic intervention.

### 5.4. Phase Separation-Related Immunotherapy

PD-1/PD-L1 immunotherapy is a highly regarded anticancer immunotherapy that uses the body’s own immune system to combat cancer. By preventing the PD-1/PD-L1 signaling pathway to cause cancer cell death, it has the potential to treat many types of tumors and is a key direction in tumor immunotherapy. The Hippo signaling pathway and its downstream effector YAP are widely involved in cancer development, progression, and therapeutic response. One study found that subcutaneous transplanted tumors of lung cancer cells developed adaptive resistance after treatment with anti-PD-1 antibodies and that YAP showed intranuclear punctate aggregation during the resistance phase [50]. Blockade of PD-1/PD-L1 activates T-cells and is accompanied by IFN-γ secretion. It was demonstrated by further studies that IFN-γ increases YAP nuclear translocation and phase separation after anti-PD-1 treatment of cancer cells. In addition, Coiled coil, a superhelical structural domain of YAP, was found to be essential for YAP phase segregation, and deletion or point mutation (4LE) of the Coiled coil structural domain could completely disrupt YAP phase segregation. YAP recruits transcription factors through phase segregation to form transcriptional hubs to maximize target gene transcription and thus efficiently complete target gene transcription [50]. In contrast, damage of YAP phase segregation reduced cancer weight, enhanced immune responsiveness, and made tumor cells more sensitive to anti-PD-1 treatment, suggesting that YAP may serve as a predictive biomarker and target for anti-PD-1 combination therapy.

### 5.5. Phase Separation Improves Peptide Drug Internalization

Self-assembly of the phosphopeptide KYp leads to phase separation at the cell membrane, thus improving cell membrane permeation and internalization of the peptide drug [149]. KYp exists as a stable single chain at pH 7.4, and when reacted with alkaline phosphatase (ALP), hydrolysis of the phosphate group leads to increased hydrophobicity, resulting in a self-assembled complex. KYp can interplay with ALP to dephosphorylate and self-assemble in situ at the cell membrane, inducing ALP aggregation and proteolipid phase separation at the cell membrane, enhancing cell membrane leakage and allowing the peptide to bypass the lysosome, resulting in a remarkable efficiency of internalization of peptide drug to reach the tumor site, and hence improving the anticancer effect [149]. Therefore, self-assembly-induced cell membrane phase separation in vivo is expected to be a novel concept to improve the effectiveness of tumor drug delivery.

### 5.6. Phase Separation Improves Drug Resistance Resensitivity and Metastasis Inhibition

The core regulatory circuitry (CRC), composed of SE-regulated core transcription factors, drives the expression of cell type-specific genes, which in turn determine cellular properties and functions. CRC in highly metastatic and chemoresistant osteosarcomas contain the important core transcription factors homeobox B8 (HOXB8) and fos-like antigen-1 (FOSL1), which produce dense dynamic phase separation droplets in vitro and phase separation agglutinates in the nucleus as well [150]. Disruption of HOXB8 and FOSL1 phase separation alters the accessibility of chromatin in the downstream SE region of CRC and inhibits the release of RNA polymerase II from the downstream gene promoter region during transcription, resulting in a blockage of the aberrant tumor transcriptional program and ultimately significantly reducing osteosarcoma growth and metastasis. Moreover, in a patient-derived xenograft model, pharmacological inhibition of CRC phase separation led to metastasis suppression and resensitivity to chemotherapeutic agents [150]. In summary, the study revealed previously unknown pharmacological strategies to specifically interfere with CRC phase separation in osteosarcoma, offering a theoretical basis for the development of novel diagnostic and treatment options for metastatic and chemoresistant osteosarcoma.

## 6. Conclusions and Perspectives

Biomolecular condensates are located in cells and consist of concentrated proteins and RNAs that are formed through a process of phase separation. These condensates play an important role in health and disease. Abnormal phase separation frequently affects the assembly and disassembly of these biomolecular condensates and is a new consideration in tumor biology that is critical for tumor development. With new research findings, more and more cancer-associated proteins have been shown to be related to phase separation, producing abnormal condensates that affect their own function and the life course of cancer cells, thus promoting cancer development.

To date, an increasing number of cancer-associated proteins have been identified and characterized, and modulation by phase separation may affect their cellular structure and function and hence influence the life course of cancer cells. Therefore, targeting cancer therapy by modulating phase separation is no longer unattainable. In this paper, we reviewed the biological properties of phase separation as well as the molecular characterization, dynamics, regulation, and function of protein phase separation. Moreover, we discussed the novel concept of abnormal biomolecular condensate driven by gene mutations and fusions in cancer. Therapeutic strategies that modulate the kinetics of cellular phase separation may be potential tools for the treatment of cancers with aberrant biomolecular cohesion. Because phase separation is a highly complex process, numerous classes of drugs, such as small molecules, antibodies targeting disordered regions, and designer peptides, may regulate the phase separation process and effectively target cancer.

The new understanding that drug design and development cells are organized into phase separated compartments, which concentrate not only cellular components but also therapeutic drugs, suggests a number of ways that may facilitate the development of novel small molecule therapies. Achieving this goal could improve the therapeutic index of the drug, resulting in lower doses and fewer side effects for patients. In addition, comprehending how mutations in protein sequences affect 3D protein composition provides mechanistic hypotheses of disease causation as well as structural medicinal chemistry approaches that can lead to valuable therapeutic outcomes. Thus, targeted cancer therapy by modulating phase separation is no longer impossible.

Undeniably, phase separation is a novel direction of biological research, and our understanding of phase separation is still in its infancy. Many questions remain about how to accurately regulate phase separation, such as how we can cleanly disrupt phase separation without altering the functions of other core proteins, how to appraise phase separation in tumor tissues, how to adjust phase separation to obtain the expected treatment impact, and how to truly know the effect of any mutation, other than altering the phase separation behavior, on protein interactions. In general, while the field of phase separation is still young, it has evolved rapidly in recent years, and the field has certainly revolutionized our cognition of diverse biological activities and disease situations. It is expected that basic research in phase separation and human disease will continue to advance as a means to identify additional potential targeting modulators that hold promise in clinical practice to help treat cancer-related diseases.

## Figures and Tables

**Figure 1 cancers-14-05971-f001:**
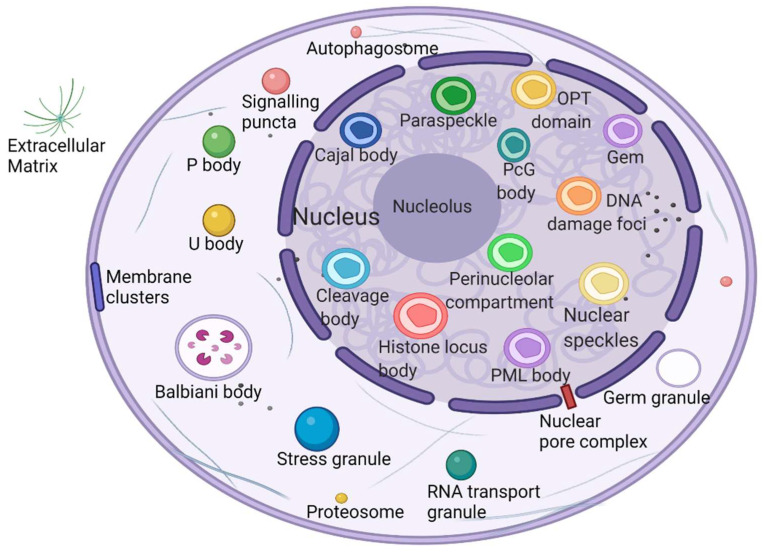
Biomolecular condensates located throughout the nucleus and cytoplasm. Created with BioRender.com.

**Figure 2 cancers-14-05971-f002:**
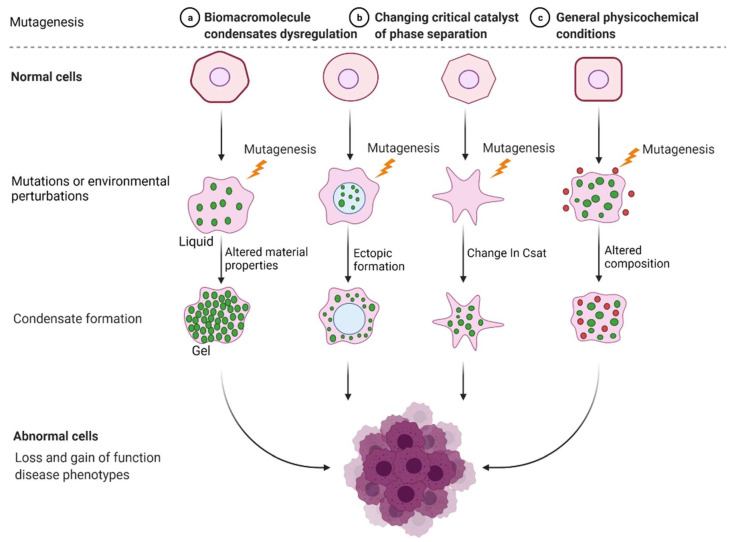
Mechanisms of abnormal phase separation in disease. Theoretical possibilities of how disease phenotypes arise from abnormal phase separation and condensate formation. Created with BioRender.com.

**Figure 3 cancers-14-05971-f003:**
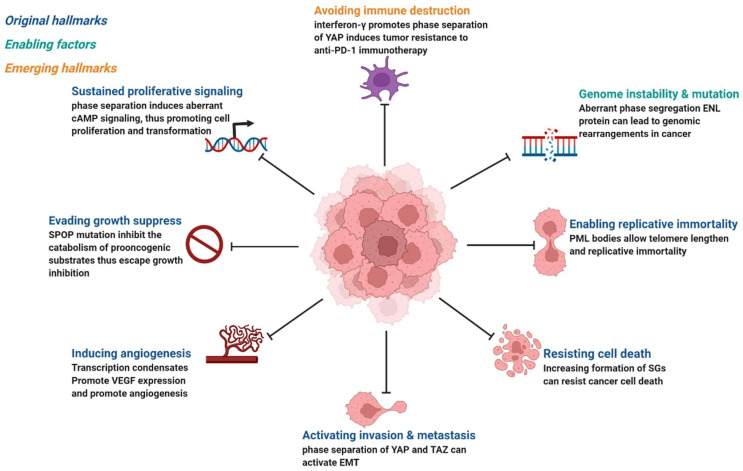
Phase separation abnormalities are involved in most of the processes known as cancer hallmarks. Created with BioRender.com.

**Figure 4 cancers-14-05971-f004:**
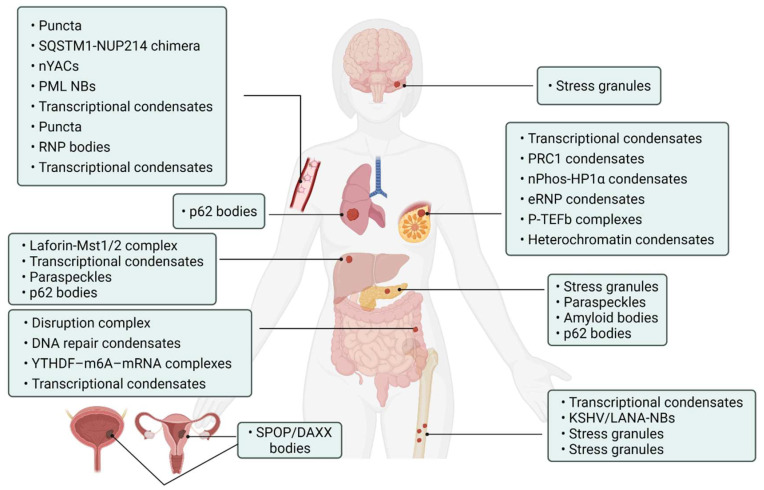
Cancer-related condensates in human cancers. Distribution of various types of cancer and related condensates in the human body. Created with BioRender.com.

**Table 1 cancers-14-05971-t001:** Condensates in cancer. Abnormal protein phase separations are involved in the progression of various cancers.

Tumor Types	Proteins	Biomolecular Condensates	Biological Roles	References
Hepatocellular Carcinoma	YAP	Laforin-Mst1/2 complex	Block Hippo kinase and accelerate tumorigenesis	[60]
	YAP, TAZ	Transcriptional condensates	Activate prevalently in cancer	[61,62]
	NEAT1_2	Paraspeckles	Induce transcription of various gene sustained by cancer cells	[63]
	p62	p62 bodies	Induce carcinogenesis	[64]
Lung Cancer	KEAP1/NRF2/p62	p62 bodies	Increase the risk of tumor genesis	[65]
Pancreatic Cancer	KRAS	Stress granules	Improve cancer cell suitability	[52,66]
	p53	Paraspeckles	Promote the expression of tumor suppressors	[67]
	ACM	Amyloid bodies	Promote tumor tissue growth	[68]
	p62	p62 bodies	Critical in regulating tumorigenesis through autophagy	[69,70,71]
Colorectal Cancer	APC	Disruption complex	Effective β-catenin degradation	[72,73]
	53BP1	DNA repair condensates	Respond to DNA damage	[74]
	YTHDF1/2/3	YTHDF–m6A–mRNA complexes	Weaken mRNA translation	[75,76]
	β-catenin	Transcriptional condensates	Wnt factor driving cancer	[77,78]
Leukemia	NUP98 FOs	Puncta	Associated with malignant transformation of hematopoietic cells	[79]
	NUP214	SQSTM1-NUP214 chimera	Associated with malignant transformation of hematopoietic cells	[80]
	YTHDC1	nYACs	Maintains mRNA stability and controls cancer cell survival and differentiation	[81]
	PML/RARA	PML NBs	Involved in oncogenic signaling	[82,83]
	MYB	Transcriptional condensates	Drive oncogenic TAL1 expression	[84]
	ENL	Puncta	Regulates oncogenic transcriptional program	[53,85]
	NPM1	RNP bodies	Ribosome biosynthesis	[86,87]
Leukemia/Sarcoma	FUS/TAF15 PLD	Transcriptional condensates	Drive aberrant tumorigenic transcriptional program	[88]
Sarcoma	KSHV/LANA	KSHV/LANA-NBs	Cause alterations in gene expression	[89,90]
	FUS/CHOP	Stress granules	Carcinogenic transformation	[91]
	EWS/FLI1	Transcriptional condensates	Promote gene transcription associated with Ewing’s sarcoma	[92]
	YB-1	Stress granules	Cancer metastatic marker	[93]
Medulloblastoma	DDX3X	Stress granules	Impair global translation	[94,95]
Breast Cancer	YAP/TAZ	Transcriptional condensates	Promote expression of target gene	[51,96]
	CBX2	PRC1 condensates	Gene suppression	[97,98,99]
	HP1α	nPhos-HP1α condensates	Epigenetic regulation	[18,19,100]
	ER	eRNP condensates	Synergistic assembly of activated chromosome enhancers	[101]
	P-TEFb	P-TEFb complexes	Activate and increase transcription of EMT transcription factors	[102,103]
	MeCP2	Heterochromatin condensates	Chromosome maintenance and transcriptional silence	[104,105]
Prostate/Endometrial cancer	SPOP	SPOP/DAXX bodies	Promote tumor development	[55,56]
Other cancers	PARP-1	DNA damage condensates	Promote DNA damage	[106,107]
	OCT4	OCT4-MED1-IDR complex	Control gene transcription	[17,108]
	MED1, BRD4	Transcriptional condensates	Activate gene transcription	[109,110,111]
	CDK7	Transcriptional condensates	Kinase overexpression and targeting in cancer	[112,113]
	HSF1	Transcriptional condensates	Act as “sensors” regulating cell fate	[114,115]
	Rad52	Repair center condensates	DNA repair	[116]
	hnRPNA1, FUS, G3NP1/2	Stress granules	Modulate the stress response	[117,118]
	FMRP/CAPRIN1	FMRP-CAPRIN1 condensates	Control RNA process and translation	[119]

**Table 2 cancers-14-05971-t002:** Abnormal biomolecular condensates in signaling pathways.

Signaling Pathways	Signaling Condensates	Effect of Phase Separation	References
cAMP/PKA	RIα condensates	Promote cell proliferation and transformation	[124]
cGAS/STING	NF2m-IRF3 condensates	Regulate tumor immunity	[125]
cEGFR/RAS	EGFR condensates	Regulate pro-tumor activation of Ras	[126,127]
Wnt/β-catenin	Destruction complex	Regulate development and stemness	[128]
RAS/MAPK	SHP2 condensates	Enhance the resistance of cancer cells to apoptosis	[59,129]
Hippo/YAP	YAP/TEAD transcriptional condensates	Act as signaling hubs for the tumor microenvironment	[130]
	Laforin-Mst1/2 condensates	Increase hepatocarcinogenesis	[60]
NRF2/NF-κB	p62 bodies	Accelerate cancer development	[65,70]

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
