# Peer review of "Protein Phase Separation: New Insights into Carcinogenesis"

_cancers, 2022, doi:10.3390/cancers14235971_

Round 1
Reviewer 1 Report
This review manuscript is timely good and well-written about emerging and important issues in biological science. Phase-separation issue regarding cancer is very important issues for recent medical studies. This review comprehensively addresses this emerging issue that can trigger many interests on potential readers.
1. This review manuscript is motivated by the importance of protein phase separation in carcinogenesis. However, in many cases, a dysregulation of phase separation come along with a dysregulation of a signaling protein which participates in the phase separation. In this regard, it would be great to have some examples that show a dysregulation driven by phase separation itself. (1) as an example.
2. Phase-separation mechanism is mainly explained by multivalent interaction between IDRs, but there is also a different mechanism of phase separation. It would be good to introduce this mechanism, called the Bridging-Induced Phase Separation (BIPS). Nowadays, this mechanism has been shown in many chromatin-related phase separation phenomena (2).
3. Although clustering of membrane proteins relates to phase separation, the evidence how EGFR’s clustering is a phase separation phenomenon does not seem to be well connected in section 4.3. The authors need to show more evidence why they believe that the clustering can be a phase-separation phenomenon.
4. Quantitative information would be preferrable than qualitative information.
(e.g., Page 1: Cancer is the main cause of death for most patients, and most malignant tumors. à Some people could argue that most death could be septic shock, etc.)
5. The authors need to explain potential strategy how to cure patients with somewhat concrete ideas in section 5.6. Otherwise, this section would be a naive idea.
1. S. Kilic, et al., Phase separation of 53 BP 1 determines liquid‐like behavior of DNA repair compartments . EMBO J 38 (2019).
2. Ryu, J., Hwang, D., & Choi, J. (2021). Current Understanding of Molecular Phase Separation in Chromosomes. International Journal of Molecular Sciences, 22(19), 10736.
Reviewer 2 Report
In this manuscript, the authors introduced the main processes, characteristics and detecting methods for phase separation. The future applications of targeted phase separation in cancer treatment are also discussed. I think this work can be accepted after minor modifications.
1. In section 2, it is interesting to know more about the role of phase separation in the normal functioning of cells.
2. In the conclusion, the authors should give more discussion about the challenges and outlooks of targeted phase separation.
